# Mechanical Properties and Functions of Elastin: An Overview

**DOI:** 10.3390/biom13030574

**Published:** 2023-03-22

**Authors:** Hanna Trębacz, Angelika Barzycka

**Affiliations:** Department of Biophysics, Medical University of Lublin, Al. Racławickie 1, 20-059 Lublin, Poland

**Keywords:** elastic fiber, elastic recoil, mechanical properties, soft tissues

## Abstract

Human tissues must be elastic, much like other materials that work under continuous loads without losing functionality. The elasticity of tissues is provided by elastin, a unique protein of the extracellular matrix (ECM) of mammals. Its function is to endow soft tissues with low stiffness, high and fully reversible extensibility, and efficient elastic–energy storage. Depending on the mechanical functions, the amount and distribution of elastin-rich elastic fibers vary between and within tissues and organs. The article presents a concise overview of the mechanical properties of elastin and its role in the elasticity of soft tissues. Both the occurrence of elastin and the relationship between its spatial arrangement and mechanical functions in a given tissue or organ are overviewed. As elastin in tissues occurs only in the form of elastic fibers, the current state of knowledge about their mechanical characteristics, as well as certain aspects of degradation of these fibers and their mechanical performance, is presented. The overview also outlines the latest understanding of the molecular basis of unique physical characteristics of elastin and, in particular, the origin of the driving force of elastic recoil after stretching.

## 1. Introduction

The mechanical role of tissues is to deliver an appropriate physical response to forces, both resulting from organ physiology and being due to external loads the body is subjected to. In order to ensure an optimal response, each tissue with defined mechanical functions should provide sufficiently strong structural support in addition to being properly deformable. The reversible deformability of the extracellular matrix (ECM) of tissues is essential for the functioning of many organs, including the lungs, skin, and blood vessels.

In a physical meaning, any material is elastic if it is able to return to its original shape and size after deformation when the force of deformation is removed. Human tissues must be elastic like all other materials designed to work under a load for a long time and without losing functionality. Although the physical meaning of elasticity does not imply high deformability, soft tissues are both elastic and stretchy, i.e., they can be largely deformed when a little force has been exerted. This elastomeric elasticity is provided by elastin, the only protein possessing this feature in mammals [1,2]. Elastin also has the ability to store elastic–strain energy with almost perfect efficiency and is extremely durable [3,4,5,6].

Considering elastin’s unique mechanical properties crucial for many vital functions of human tissues and organs and its role in various biological mechanisms [4,7,8], it is not surprising that there is a lot of interest in elastin in many research areas. Review papers on elastin in the context of its biology and biochemistry [4,9,10,11,12,13], mechanical functions [1,5], diseases, and aging [14,15,16,17,18,19,20] are widely cited, and new data are constantly emerging. Another important issue is the huge potential of elastin and elastin-like peptides in biomedical applications, including advanced biomaterials and regenerative medicine [21,22,23,24,25].

The structure of elastin and the molecular mechanism of its elasticity has been a matter of debate for several decades [26,27,28,29,30,31,32,33,34,35,36,37]. While there was a wide consensus on the entropic origin of elastin elasticity, the main difference among the models was the presence and nature of the ordered structures that contribute to the molecule entropy.

This work aims to provide a concise overview of the mechanical properties of elastin, its role in tissue elasticity, and current knowledge on the molecular basis of elastin’s unique physical performance.

## 2. Elasticity of Soft Tissues

The description of the mechanical behavior of tissues requires certain physical parameters that can be used to assess the response of tissues to applied loads. As with other materials, the parameters that quantify the mechanical properties of the tissue are based on the relationship between the forces acting and the result of these forces expressed in terms of shape changes, resistance to deformation, and the energy involved in this process. Extensibility, modulus of elasticity, elastic–strain energy, and ultimate strength can be derived from the relationship between the force applied to the material being stretched and the resulting extension, which is expressed in the form of stress–strain curve. The resistance of the material against deformation when subjected to a given stress, that is, its stiffness is expressed as Young’s modulus (modulus of elasticity) and is calculated as the slope of the stress–strain curve within the linear region where the material deforms fully reversibly. The higher Young’s modulus, the stiffer material and the greater its ability to transmit forces and resist deformation [1,5,38]. Thus, a more deformable compliant tissue will exhibit a lower elastic modulus than a less deformable “stiffer” one. As concerns the energy absorbed by the material during deformation, elasticity implies that it will be recovered during recoil [1,5,39]. The efficiency of energy recovery in a deformation: the recoil cycle is expressed as resilience, which should be 100% in a fully elastic deformation.

The mechanical functionality of soft tissues is provided by fibrous components of extracellular matrix (ECM), collagen, and elastic fibers. Collagen is the most abundant component of the tissues ECM. It is responsible for tensile strength and plays a crucial structural role [1,5,38,40]. A variety of collagen types give rise to an impressive diversity of three-dimensional supramolecular structures compatible with tissues’ mechanical functions and the forces the tissues must handle [41,42,43]. The main component of elastic fibers, elastin, has a low modulus of elasticity and deforms reversibly with very high resilience. The key function of elastin in ECM is to provide low stiffness, high extension and efficient elastic–energy storage [1,4,5]. Although much less abundant than collagen, elastin is present in large amounts within highly elastic tissues like arteries and lungs, where repetitive extensions and relaxations are essential for their function [44,45]. Although mature collagen and elastin networks function in the same tightly filled extracellular matrix, they remain structurally independent of each other. Very few physical interactions between collagen and elastin in the ECM have been documented [9]. However, the coexistence and synergy of collagen and elastin networks result in the nonlinear elastic response of tissues. A typical stress–strain curve for tissue samples is not linear but J-shaped, where initial response at low extension is due to compliant elastin, whereas, at higher extensions, the loads are transferred by stiffer collagen, so elastic stiffness of tissue increases with loading [1,2,38,39,40]. This may cause some ambiguity when attempting to quantify tissue elasticity, as the modulus of elasticity is the function of strain.

Another cause of nonlinearity in the stress–strain relationship is the viscosity of tissue components. Both fibrous protein networks are immersed in a water-saturated, viscous milieu of ECM rich in glycoproteins, proteoglycans (PGs), and glycosaminoglycans (GAGs) [3,5,46]. These viscous liquid components make the tissues not perfectly elastic. Tissues typically exhibit viscoelastic behavior, which is due to the fact that the reaction of fibrous components to tissue deformation in a viscous environment is time-dependent. Moreover, the interactions between the elastic and viscous components observed in each deformation–recoil cycle result in a dissipation of a certain amount of strain energy as heat, and consequently, the elasticity and resilience of the tissues are never 100% [1,5,39,40].

The characteristic nonlinear nature of the stress–strain relationship with the deformation-dependent modulus of elasticity and not perfect energy recovery is similar for different tissues; however, their functional and ultimate strains, moduli of elasticity, and ultimate strength differ greatly [21,47,48,49,50,51]. Table 1 presents some examples of soft tissues’ mechanical characteristics. It should be underlined that the diversity of experimental data is affected not only by the type of sample and its location in the tissue but also by the type and parameters of the mechanical test performed. Moreover, determining the modulus of elasticity in viscoelastic material may be ambiguous.

### 2.1. Occurrence of Elastin in Tissues

As different types of tissues exhibit different mechanical functions and requirements for elasticity, the content and arrangement of elastin vary between and within tissues. Elastic fibers are mostly present in elastic tissues such as the blood vessels and lungs, where their architecture and mechanical role is well-understood and frequently described [15,18,20,39,40,44,45,52,53]. Moreover, in the skin where elastin is present in small amounts, its profound impact on mechanical behavior has been known for many years [54]. As was discussed in detail by Green et al. [2], elastin is a more widely distributed component of tissues than was previously supposed, and elastin fibers approximately 1 µm in diameter are common building blocks forming the elastic structures of many tissues. Developing microscopic techniques have allowed the revealing of complex networks of fine elastin fibers in other tissues such as small blood vessels, cartilage, intervertebral discs, and even in the adipose tissue and tendons [2]. Table 2 gives an overview of elastin amounts obtained from tissues’ dry weight: from 70% in nuchal elastic ligaments to less than one percent in the meniscal fibrous cartilage.

The role of elastin in tissue elasticity results not only from its amount but also from the spatial arrangement and the type of network being created. One of the richest sources of elastin is the nuchal ligament, where elastin forms a filamentous network that orients itself parallel to the direction of stretching, along the spinal cord, providing head support to large mammals [5]. In the elastic arteries, particularly in the aortic wall where elastin is the major component, elastin fibers surrounded by circumferentially oriented smooth muscle cells and collagen fibers form highly organized and thick concentric lamellae. Such an arrangement lets the artery diameter follow changes in blood pressure and hemodynamic stresses during the cardiac cycle [20,64]. Moreover, physical connections and synergy between elastic fibers and muscle cells ensure the proper response of the cells to mechanical strain [65]. In the muscular arteries, where elastin fibers are also abundant, they do not form such regular lamellar units as in elastic ones [66]. Elastin in small resistance arteries forms a thin layer of longitudinally aligned fibers in media, while in the adventitia, the fibers are more abundant and create a more complex network [2]. Longitudinally aligned adventitial elastin fibers were found in arterioles subjected to longitudinal stretch [67].

Generally, the structure and the amount of elastin in different types of blood vessels depends on their location in the circulatory system and local hemodynamic conditions. The contribution of elastic structures to vascular biomechanics has been exhaustively studied for decades [18,20,40,44,53,68], but new papers are constantly being published [15,65,69,70,71,72,73,74,75] showing that there is still room for exploration in this area.

A complex alignment of elastin fibers can be found in the heart valves. Different layers of the heart valve have different mechanical properties owing to the amount and arrangement of elastin present in each layer [53]. Although they do not contribute to stiffness and strength, the elastic fibers provide flexibility and stretch in response to the hemodynamic environment and significantly contribute to valve performance during the cardiac cycle [76]. A continuous mechanical efficiency of the valves is an example of perfect mechanical cooperation between collagen and elastin within tissues ECM.

Elastin is widely distributed in the lung compartments, with the highest concentration in the respiratory parenchyma [52]. The lung elastic fibers exhibit significant structural heterogeneity, and the distribution of diameters and lengths of the fibers appear similar to the distribution of collagen fibers [45].

Although present in much smaller amounts, tiny elastin fibers also have a big impact on the properties of other tissues. The studies on the articular cartilage showed a fine and dense network of elastin fibers located around the chondrocytes [2,77]. They seem to protect the cells from moderate stretching forces spreading inside the loaded cartilage. At the superficial layer, where the higher tensile strains are present, a cobweb-like elastin fiber network is observed, which increases the resistance of the cartilage to strain in different directions [77]. In the intervertebral disc, the multi-scale hierarchical structure of the elastic fibers plays a significant biomechanical role [78,79]. The organization of the fibers located between the lamellae of collagen fibers changes along with the load exerted on the disc [80], age, and pathology [79,81]. In the skin, the elastic fibers form a three-dimensional meshwork that spans from the papillary down to the deep dermis and surrounds densely packed collagen fibers [62,82,83]. The meshwork consists of branched fibers of extremely variable width [62]. The mechanical response of the tissue is determined by the amount and spatial arrangement of elastin and collagen fibers relative to each other [83]. Elastic fibers in the collateral ligament are oriented along the collagen fibers but form an isotropic matrix in the transverse plane providing resistance to multiaxial deformations [84].

### 2.2. Mechanical Performance of Elastic Fiber

The ability of diverse tissues with different functional mechanical requirements to deform and to effectively regain their shape after deformation is provided by the same elastin-rich structure of elastic fibers [9,10]. Elastic fibers can be linearly extended more than twice their length before rupture occurs, and once tension is released, they return to their original dimensions without hysteresis [1].

The mammalian elastic fibers consist of an inner cross-linked and insoluble elastin core (90% of the volume) surrounded by a shell of tiny microfibrils (10–12 nm in diameter), which are two orders of magnitude smaller than the diameter of the entire elastic fiber [5,14,85]. The complex process of elastic fiber formation is regulated at multiple steps, including coacervation, deposition, cross-linking, and assembly of insoluble elastin onto indispensable microfibril scaffolds [4,5,8,9,13,14,86,87]. Fibrillin is the major component of microfibrils. There are species- and tissue-dependent differences in the expression levels of three isotypes of fibrillin, with fibrillin-1 being the predominant isotype found in adult human tissues [9,88,89,90]. Although fibrillin is the major component of the microfibrils, an array of other less abundant molecules is necessary, including latent transforming growth factor-β binding proteins (LTBPs), matrix-associated glycoproteins (MAGPs), members of the fibulin family, and PGs. They play an essential role in the organization of fibers, but also their complex hierarchical assembly supports the biological functions of microfibrils, including induction of cellular responses to mechanical forces derived from the matrix microenvironment [10,14,87,91]. Therefore, the two basic components of an elastic fiber have distinct tasks in the tissue ECM; elastin stores energy of deformation and provides passive recoil, whilst fibrillin microfibrils direct elastogenesis, mediate cell signaling, and maintain tissue homeostasis [9,11,14,89,90].

Young’s modulus of elastic fiber-rich samples from purified arteries ranged at 0.13–0.65 MPa in dog and sheep aorta [92] and between 0.1 and 0.8 MPa in pig aorta [93]. Young’s modulus of single elastic fibers isolated from bovine nuchal ligament ranged at 0.4–1.2 MPa [26]. According to Koenders et al. report [85], it was within the range of 0.3–1.5 MPa.

So, the stiffness of elastin expressed in terms of Young’s modulus is at least two orders of magnitude smaller than that of collagen [1,39]. In general, low Young’s modulus is due to the high extensibility of the material, and in fact, the maximum elastic elongation of elastin exceeds 100% [1]. The breaking strain was reported to be up to 200% [26].

Another interesting issue is whether the mechanical properties of tissues result from elastin molecules or do fibrillin–microfibrils play a role. The contribution of microfibrils to the mechanical performance of the meshwork of elastic tissue was experimentally studied in pig aorta [93] and in bovine nuchal ligament [85]. Removal of the microfibrils from elastic fibers from the pig aorta reduced the modulus at low strains by a few percent and increased the modulus at high strains, suggesting that the microfibrils have the capacity to change the orientation of elastin fibers, possibly transmitting some of the load from one elastin fiber to another [93]. Koenders et al. [85] showed that Young’s moduli of single elastic fibers from bovine nuchal ligament were not significantly affected by the absence or presence of fibrillin–microfibrils. Young’s modulus for pure fibrillin–microfibrils ranged from 0.56–0.74 MPa, which was comparable with Young’s modulus of elastic fibers cited above. They concluded that fibrillin–microfibrils did not significantly influence the mechanical properties of single elastic fibers in the vertebrate. However, Sherratt et al. [94], based on the linear springs model of microfibrils, estimated Young’s modulus of single fibrillin–microfibrils from zonular filaments of the eye to be 78–96 MPa, which was two orders of magnitude higher than the modulus determined for elastic tissue samples. The authors suggested that microfibrils have their own mechanical role in elastic fibers and act as relatively stiff reinforcing components in a fibrous composite. Megill et al. [95] showed that reinterpretation of the data presented by Sherratt et al. [94] in terms of the nonlinear model of mechanical behavior should result in at least one order of magnitude lower value. They suggested Young’s modulus of 1 MPa for fibrillin–microfibrils in a fiber-reinforced composite model.

### 2.3. Driving Force of Elastic Recoil

To understand the origin of the mechanical efficiency of tissues both in a healthy and diseased or/and aging organism, we must remember that the basis for elasticity and resilience of tissues is an exceptional capability of cross-linked elastin fibers to extreme deformation under small loads and next spontaneous recoil back to the original shape with minimal energy loss. In all materials, also in living tissues, elastic recoil after deformation results from the sum of two different physical driving forces. One of them appears as a reaction to internal energy changes when the applied deforming force distorts the molecular structure of the material and results from the tendency of each molecular system to a spontaneous regain of the state of the lowest potential energy. The other one results from a thermodynamic principle stating that isolated systems spontaneously arrive at a state where entropy is the highest in given circumstances. The first process dominates in stiff materials with ordered molecular structure, while the other one is in elastomers, elastically deformable polymers characterized by a high degree of conformational disorder which makes the elastomers in a relaxed state have high entropy [6,96]. There are two essential molecular attributes of material with entropic elasticity: flexible polymer chains and the presence of cross-links between them. Elastin structure is characterized by a high degree of conformational disorder, which makes it flexible and easily stretched, and by a high degree of cross-linking, resulting in a network capable of distributing the deformation-related stresses and strains throughout the polymer. The entropic component of elastin elasticity is more than 70% [96]. Thus a passive, entropy-driven mechanism allowing the recoil of elastic fibers after stretching endows the extracellular matrix of connective tissues with their elasticity and resilience.

Various models of entropic elasticity have been proposed, ranging from maximally disordered isotropic structure [28,29] to highly organized arrays of beta-spirals and beta-turns [33], and the main driving force of elastic recoil has been sought either in the conformational entropy of polypeptide chains [29] or in the entropy of librational motions of fixed secondary structures suspended between mobile chain segments [33] or in the hydrophobic effect [28,30]. The current consensus is based on a model in which the water-swollen hydrophobic domains of elastin molecules form highly disordered but not a random assembly of dynamic conformations devoid of permanent secondary structures [31,32,35,36,37].

### 2.4. Molecular Basis of Elastin Elasticity

In all tissues, no matter how different they are, the basis of elasticity is the same and is “encoded” in the molecular structure of elastin. Human elastin is secreted principally from fibroblasts and smooth muscle cells as tropoelastin, a highly hydrophobic, ~60 kDa unglycosylated monomer. The primary sequence of tropoelastin is formed by an arrangement of two major types of alternating domains, the hydrophobic domains and the hydrophilic lysine-rich domains [97,98,99]. The hydrophobic domains are rich in glycine, proline, and valine, commonly arranged in combinations of GV, GVA, and PGV sequences. The hydrophilic helical domains of tropoelastin contain lysine residues spaced three or four residues apart and typically flanked by alanines. In the extracellular space, the tropoelastin units are chaperoned to the cell surface, where they coacervate into protein-rich spherules and then undergo cross-linking and fibril assembly onto microfibril scaffolds, which has been described many times and is updated every few years [8,10,13,86,87,99,100,101,102,103]. The cross-linking of tropoelastin monomers occurs via lysine residues present in helical domains through the action of lysyl oxidases, which results in the formation of tetra-functional desmosine and isodesmosine linkages and bi-functional allysine–aldol and lysinonorleucines. Cross-linking between hydrophilic helical domains of elastin monomers stabilizes elastin microfibrils and provides elastic fibers with structural integrity and durability, and contributes to their high insolubility. Recent reviews of elastin cross-links biochemistry are available in refs. [8,87,104,105].

While elastin’s high structural integrity and durability are due to cross-links formed in its hydrophilic domains, its elasticity results from the specific sequences of hydrophobic amino acids. However, despite the strong hydrophobicity of the elastin monomer containing ~80% non-polar amino acids in its structure, hydration of elastin is an absolute requirement for elasticity [1,30,37,97]. Dry elastin is hard and brittle, while elastin monomers in a water environment are disordered and flexible [31,37]. They retain backbone mobility even after aggregation [31] and in mature cross-linked fibers [36].

In general, proteins composed of non-polar amino acids tend to form tightly packed and ordered secondary structures shielding non-polar side chains from the surrounding polar environment. In the elastin monomer surrounded by water, the formation of an ordered secondary structure is prevented by high glycine and proline content, which accounts for 30% and 12% of its amino acids, respectively [31]. The fixed ϕ dihedral angle and lack of amide hydrogen of proline, as well as the flexibility of small glycine, make their sequences prevent the formation of a compact, water-excluding core, which maintains a high degree of structural disorder and allows water molecules to spread among elastin network [31]. The solvent water molecules act on elastin as a plasticizer by interacting with water bound to the main chain, which allows the chain to be more mobile [30]. Although the sequences of hydrophobic amino acids prevent the formation of large secondary structures, the dynamics of the hydrated backbone of the elastin molecule results in transient hydrogen-bonded turns, which form highly disordered, but not random, assemblies of dynamic conformations such as short and labile beta structures and polyproline II helices [31,37]. Both molecular dynamics simulations of elastin-like peptides sequences [30,32] and solid-state NMR experiments with mature elastin [36,106] reveal the extremely dynamic nature of hydrophobic domains providing high entropy of elastin in the relaxed state (Figure 1, left side). Rausches and Pomes [32], based on massive-scale molecular dynamics simulations, describe the assembly of elastin individual chains in water as a maximally disordered, melt-like state—a liquid state of elastin. However, it has been shown that also hydrophilic, alanine-rich cross-linking domains can significantly contribute to complex elastin conformations [107].

The extension of cross-linked elastin leads to a decrease in the conformational entropy of individual chains as well as polymerized material (Figure 1, right side). Thus, both the hydrophobic effect and conformational entropy related to a high structural disorder of the polypeptide chain drives the elastic recoil of stretched elastin molecules. However, there is still a lack of full understanding of all the mechanisms underlying the extraordinary elasticity of cross-linked elastin in elastic fibers.

### 2.5. Degradation of the Elastic Fiber Mechanical Performance

Elastin expression in mammals begins in mid-gestation and continues at high levels through childhood. However, elastin synthesis after adolescence is diminished; thus, mature elastic fibers have to fulfill their biomechanical function almost over the entire life of the organism. In contrast to continuously synthesized intracellular proteins, elastic fiber proteins are remarkably long-lived, with an in vivo half-life of elastin in humans estimated to be of 70 years [9,14,108]. The composition of elastin molecules, extremely dense packing, and high degree of crosslinking make elastin the most stable of the extracellular matrix molecules [1,4,9,11]. Consequently, elastic fibers are resistant to most influences and, under normal conditions, are able to undergo billions of cycles of extension and recoil without mechanical failure. However, it is commonly observed that in aging humans, cardiovascular, pulmonary, and dermal tissues become increasingly stiff and lose their essential ability to regain shape in a fast and effective way. The loss of elasticity in the skin, blood vessels, lungs, and other tissues is an undoubted sign of the aging process. The loss of functionality of elastic fibers is due to both fragmentation and/or thinning of elastin networks as well as modifications of the elastic properties of the fibers themselves [16,17,19,72,108,109]. The effect of fibers degeneration results in an excessive transfer of mechanical loads to collagen and the consequent stiffening of tissues [1,5,20].

The progressive loss of the mechanical function of aging fibers is a consequence of a continuous accumulation of damage resulting from chemical and physical processes induced by both intrinsic and extrinsic factors. Based on the current models and reviews [8,14,16,17,19,110], the mechanisms of elastic fibers degradation can be briefly summarized as follows. The very low turnover of the fibers makes them prone to enzymatic proteolysis by the family of extracellularly acting proteinases, as well as to reactive oxygen species (ROS)-mediated oxidation, formation of advanced glycation end-products (glucose-mediated cross-linking), calcium accumulation, binding of lipids and lipid peroxidation products, carbamylation, time-dependent modification of aspartic acid residues, and mechanical fatigue. Moreover, extensive research has shown that calcification, cholesterol binding, glycation, enzymatic degradation resulting in the release of elastokines, and chronic low-grade inflammation can complement and enhance each other. It was shown that even though a healthy lifestyle can help reduce extrinsic risk factors and is able to postpone the onset of elastic fibers weakening, it cannot fully prevent intrinsic degradation processes in the extracellular matrix of aging tissues. Robert et al. [111] have estimated that an upper limit for the mechanical performance of the human cardiorespiratory system is about 100–120 years. However, intensification and/or coexistence of degradative mechanisms can cause severe pathologies involving the cardiovascular system, skin and lungs much earlier, not only at an advanced age [8,14,19,110].

Though the gradual degradation of elastic fiber and associated dysfunctions result inevitably from multiple physiological intrinsic processes and some extrinsic factors impairing tissue homeostasis, a certain number of inherited elastic-fiber pathologies were also recognized. They result from mutations in the genes encoding elastin, fibrillin, and/or other proteins involved in microfibril and elastic fiber assembly. Marfan syndrome is the most frequent genetic disease directly associated with mutations in the fibrillin genes [14]. A detailed list and description of mutations and symptoms in inherited elastin- and elastic-fibers pathologies can be found in a detailed review by Baldwin et al. [14] and more recent ones [110,112].

## 3. Conclusions

The aim of this paper was to overview the literature regarding the role of elastin in the elasticity and mechanical performance of tissues as well as a brief description of the molecular model of elastin’s unique properties.

Elastin is a unique long-living molecule that works as strain–energy storage and provides vertebrate tissues with the extensibility and elasticity necessary for the functioning of vital organs. Different mechanical requirements of various tissues are met by the same elastin-rich structure, i.e., elastic fiber. The variety of mechanical functions and resulting mechanical parameters of tissues are provided by different amounts of elastic fibers, their arrangement, and mechanical cooperation with other components of the tissue extracellular matrix. Moreover, the analysis of experimental data referring to tissues should also account for other factors, such as the nonlinear strain–stress relationship in viscoelastic materials and testing methods.

There is a consensus on the entropic origin of elastin elasticity; however, the nature of the structures that contribute to the molecule entropy is continually being studied. In the currently accepted model, high proline and glycine content in the elastin monomer prevents the formation of compact hydrophobic structures and allows water to spread among elastin molecules. The water-swollen hydrophobic elastin domains form a highly disordered and dynamic, but not random, assembly of conformations devoid of permanent secondary structures.

Numerous significant advances in life sciences produced evidence that elastic fibers and their breakdown products, especially elastokines, are implicated in the etiology of numerous diseases as well as in aging-related health problems. Even though modern experimental and simulation methods brought a huge insight into the biology and physical interactions of elastin, some aspects of its functioning in the ECM structures, as well as mechanisms of the enormous flexibility of cross-linked elastin, remain unexplained. Further expansion of knowledge of their synthesis and decomposition would aid the development of innovative treatment methods and should hopefully lead to new strategies for elastic fiber repair and regeneration.

## Figures and Tables

**Figure 1 biomolecules-13-00574-f001:**
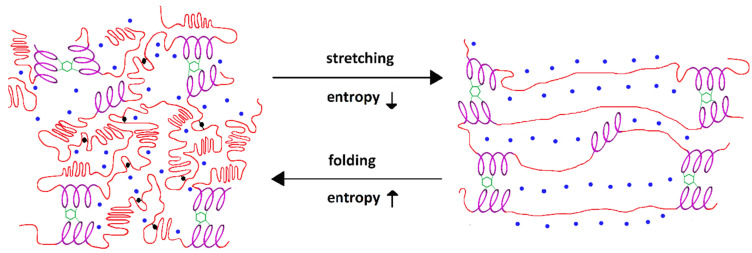
Simplified model of hydrated crosslinked elastin; helical, hydrophilic domains of polypeptide chains (magenta), cross-linking between helical domains (green), hydrophobic domains (red), peptide–peptide hydrogen bonds (black), solvating water (blue). Folded, native state (left side)—Prolines and glycines prevent the hydrophobic collapse of the hydrophobic domains, which allows water molecules to spread among the elastin network; interactions of the solvent water molecules with water bound to the main chain allow for the chain mobility and result in transient hydrogen-bonded turns as short and labile folded structures. Extended state (right side)—Extension of elastin leads to a decrease of conformational entropy of the polypeptide chains and increases hydrophobic interactions with exposed hydrophobic residues; both the hydrophobic effect and conformational entropy of the chain drive the elastic recoil of stretched elastin molecules.

**Table 1 biomolecules-13-00574-t001:** Mechanical parameters of selected mammalian tissues.

	Elastic Modulus, (MPa)	Maximal Strength, (MPa)	Maximum Strain, (%)	Refs
Elastin free tendon	1200	120	13	[1]
Elastin from nuchal ligament	1.1	2	150	[1]
Arteries and veins (different species)	0.6–3.5	2	-	[48]
Cortical artery (human)	21.4	4.1	145	[49]
Cortical vein (human)	3.4	1.4	193	[49]
Aortic valve leaflet human	15.6	2.6	21.9	[49]
Tendon (different spices)	43–1660	560		[48]
Tendon (human)	143–2310	24–112	-	[49]
Ligament (human)	65–541	13–46	-	[49]
Skin (different species)	21–39	30		[48]
Skin (rat)	25.35	7.83	46	[50]
Articular (cartilage bovine)	30	-	-	[51]
Auricular (cartilage bovine)	15	-	-	[51]

**Table 2 biomolecules-13-00574-t002:** Amounts of elastin in dry mass of human and bovine (*) tissues.

	Elastin Amount (%)	References
Nuchal ligament *	~70	[22,55]
Large arteries	>50	[10]
Yellow ligament	~47	[56]
Saphenous vein	~32	[57]
Lung parenchyma	20–30, ~30	[52,58]
Auricular cartilage *	19, 20	[51,59]
Auricular cartilage	15	[60]
Heart valves	10–15	[55]
Pulmonary blood vessels	7–16	[52]
Mitral valve chordae tendineae	~5	[61]
Airways	3–5	[52]
Skin	2–4, 3–4	[10,62]
Nasal cartilage	3–5	[60]
Intervertebral disc	1.7, 2	[56,63]
Meniscus	0.6	[59]

## Data Availability

Not applicable.

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
