# Peer review of "Mechanical Properties and Functions of Elastin: An Overview"

_biomolecules, 2023, doi:10.3390/biom13030574_

Round 1

Reviewer 1 Report

This review is based on elastin and its mechanical requeriments. It is novel and could be interesting for the readers of this journal. Nevertheless, it is necessary some minor changes before it publication:

-Table 1: Remove % in the values of maximal strain. It should be put only in the first row together with the name.

-Change "maximal" for "maximum"

-Table 2: Remove % in the values of elastin amount. It should be put only in the first row together with the name.

-The novelty and interest of this topic has been stressed in the introduction. However, most of the submitted articles (80%) are not from the last 5 years. References should be updated.

Author Response

Reviewer # 1

  • The Tables are corrected.
  • Five new articles relevant to the topic of our paper are included: #65 (p.6), #70 (p.6), #84 (p.7), #100 (p.10) and #107 (p.11)

Reviewer 2 Report

This manuscript is a concise review written by Trebacz and Barzycka that summarizes the mechanical properties of elastin in different tissues. My main comments are:

·      On page 2, lines 87-90 state that collage and elastin are independent of each other with very few interactions noted, while on page 5, lines 173-174 state that the “mechanical response of the tissue is dominated by elastin’s interactions with collagen networks. These 2 statements are conflicting. A change in the wording of the latter statement may be necessary.

·      In my humble opinion, the order of data presentation needs to be changed. For instance, section 7 “Molecular basis of elastin elasticity” should certainly go before section 5 describing “Degradation of the elastic fiber.”

·      The manuscript would benefit from editing by a native English-speaking individual.

Author Response

  • The unfortunate statement ending the paragraph “Occurrence of elastin in tissue” – “ Mechanical response of the tissue is dominated by elastin’s interactions with collagen networks” was replaced by “The mechanical response of the tissue is determined by the amount and spatial arrangement of elastin and collagen fibers relative to each other “.
  • According to the suggestions the section “Degradation of the elastic fiber” was moved to the end.
  • The manuscript was reviewed by native English-speaker.

Reviewer 3 Report

In this manuscript, anna Trębacz and Angelika Barzycka of the mechanical properties of elastin and its role in the elasticity of soft tissues. However, at this stage there are still many problems and I therefore suggest a major review for this manuscript keeping in mind the following questions.

1)The introduction is not meaningful. The respected authors are requested to write a comprehensive introduction related to the topic with concentration on the paragraphs consistent with the preceding ones.

2) For a review the conclusion is very important so that the ideas are very clear for our readers. I invite you to rewrite the conclusion and make it consistent and attracts readers.

3) The manuscript needs extensive revision for language and grammar

Author Response

  • The introduction was shortened. The paragraph referring to various models of elastin structure was removed from Introduction and incorporated into section “Driving force of elastic recoil”
  • Conclusions are rewritten.
  • The manuscript was reviewed by native English-speaker.

Round 2

Reviewer 2 Report

The authors addressed my comments for the most part. 

Minor editing, particularly for English (the use of "the" or lack thereof), is needed, otherwise I have no concerns.

Reviewer 3 Report

Accepted